# Circulating Soluble Urokinase-Type Plasminogen Activator Receptor in Obstructive Sleep Apnoea

**DOI:** 10.3390/medicina56020077

**Published:** 2020-02-14

**Authors:** Renata Marietta Bocskei, Martina Meszaros, Adam Domonkos Tarnoki, David Laszlo Tarnoki, Laszlo Kunos, Zsofia Lazar, Andras Bikov

**Affiliations:** 1Department of Pulmonology, Semmelweis University, 1083 Budapest, Hungary; drbocskeirenata@gmail.com (R.M.B.); martina.meszaros1015@gmail.com (M.M.);; 2Department of Pulmonology, Szent Borbala County Hospital, 2800 Tatabánya, Hungary; 3Medical Imaging Centre, Semmelweis University, 1082 Budapest, Hungary; tarnoki2@gmail.com (A.D.T.); tarnoki4@gmail.com (D.L.T.); 4North West Lung Centre, Manchester University NHS Foundation Trust, Manchester M23 9LT, UK

**Keywords:** biomarkers, fibrinolysis, inflammation, OSAHS, sleep disordered breathing

## Abstract

*Background and Objectives*: Obstructive sleep apnoea (OSA) is associated with heightened systemic inflammation and a hypercoagulation state. Soluble urokinase-type plasminogen activator receptor (suPAR) plays a role in fibrinolysis and systemic inflammation. However, suPAR has not been investigated in OSA. *Materials and Methods*: A total of 53 patients with OSA and 15 control volunteers participated in the study. Medical history was taken and in-hospital sleep studies were performed. Plasma suPAR levels were determined by ELISA. *Results*: There was no difference in plasma suPAR values between patients with OSA (2.198 ± 0.675 ng/mL) and control subjects (2.088 ± 0.976 ng/mL, *p* = 0.62). Neither was there any difference when patients with OSA were divided into mild (2.134 ± 0.799 ng/mL), moderate (2.274 ± 0.597 ng/mL) and severe groups (2.128 ± 0.744 ng/mL, *p* = 0.84). There was no significant correlation between plasma suPAR and indices of OSA severity, blood results or comorbidities, such as hypertension, diabetes, dyslipidaemia or cardiovascular disease. Plasma suPAR levels were higher in women when all subjects were analysed together (2.487 ± 0.683 vs. 1.895 ± 0.692 ng/mL, *p* < 0.01), and also separately in controls (2.539 ± 0.956 vs. 1.411 ± 0.534 ng/mL, *p* = 0.02) and patients (2.467 ± 0.568 vs. 1.991 ± 0.686 ng/mL, *p* < 0.01). *Conclusions*: Our results suggest that suPAR does not play a significant role in the pathophysiology of OSA. The significant gender difference needs to be considered when conducting studies on circulating suPAR.

## 1. Introduction

Obstructive sleep apnoea (OSA) is a common disease which is characterised by repetitive collapse of the upper airways during sleep which results in intermittent hypoxia and frequent microarousals. These processes lead to the development of cardiometabolic comorbidities, such as hypertension, cardiovascular disease, diabetes and dyslipidaemia, which frequently accompany OSA.

Chronic intermittent hypoxia and increased sympathetic tone induce production of pro-inflammatory molecules, such as interleukin (IL)-6, IL-1β, tumour necrosis factor-α [1] or complement elements [2] and suppresses the release of anti-inflammatory [3,4] molecules. Linked to inflammation and sympathetic activity, OSA is characterised by a hypercoagulation state [5,6,7,8,9]. Accelerated systemic inflammation and increased coagulation may contribute to the development of cardiovascular disease and acute cardiovascular events [10].

Soluble urokinase-type plasminogen activator receptor (suPAR) is a molecule which plays a role in both inflammation and coagulation. It is produced upon cleavage of the membrane-bound urokinase-type plasminogen activator receptor (uPAR). The cleavage is facilitated by urokinase-type plasminogen activator (uPA), plasmin, matrix metalloproteases, neutrophil elastase and cathepsin G [11]. The urokinase receptor (also known as uPAR) is expressed by endothelial cells, macrophages, monocytes, neutrophils, lymphocytes, smooth muscle cells and fibroblasts [12,13]. It is upregulated under infections and as an effect of pro-inflammatory cytokines [11,13,14,15], while suPAR contributes to plasminogen activation, cell adhesion, chemotaxis and immune cell activation [16]. However, uPAR also acts as a scavenger receptor for uPA, inhibiting its actions [17]. In large studies, plasma suPAR was elevated in coronary artery disease and cerebrovascular disease and correlated with their severity and cardiovascular mortality [18,19,20]. Higher suPAR levels were also observed in obesity [21], which is the main etiological factor for OSA in the Western population [22].

Only one study has investigated suPAR in probable OSA so far [23]. Patients were categorised to high and low-risk OSA based on the Berlin questionnaire and neck circumferences; however, no objective sleep tests were performed. In this study, there was a tendency for higher suPAR levels in the high-risk group, but the difference did not reach a significant level [23].

We hypothesised that circulating suPAR concentrations are elevated in OSA compared to health and probably relate to disease severity. The aim of the study was to investigate these using standardised diagnostic tests.

## 2. Materials and Methods

### 2.1. Study Design and Subjects

We recruited 68 volunteers (54 ± 13 years, 36 men) who were referred for a sleep study to the Sleep Unit, Department of Pulmonology, Semmelweis University due to suspected OSA (i.e., snoring, daytime tiredness, obesity, comorbidities). After giving informed consent, medical history was taken and patients filled out the ESS, which was followed by in-laboratory cardiorespiratory polygraphy (*n* = 20) or polysomnography (*n* = 48). In the morning, blood pressure was measured; fasting venous blood was taken for lipid profile, glucose, creatinine, C-reactive protein (CRP) and suPAR measurements between 6:00 and 8:00 a.m. Glomerular filtration rate (GFR) was calculated using the Modification of Diet in Renal Disease equation.

Comorbidities were defined according to the participants’ report, available medical records, medications, morning blood pressures and fasting blood laboratory results. In detail, hypertension was excluded if there was no history for high blood pressure. Participants did not take anti-hypertensive medications, and morning blood pressure was within the normal range. In line with this, diabetes and dyslipidaemia were excluded if there was no history for these comorbidities, participants did not take antidiabetic or anti-dyslipidaemia medications, and the fasting blood glucose and lipid results were in the normal range. Cardiovascular disease was excluded based on absence of symptoms and negative medical history.

### 2.2. Sleep Studies

Inpatient polysomnography and cardiorespiratory polygraphy were performed as described previously [2,3,4] using Somnoscreen Plus Tele PSG (Somnomedics GMBH Germany). Sleep stages, movements and cardiopulmonary events were scored manually according to the American Academy of Sleep Medicine [24] guidelines. Apnoea was defined as a 90% airflow decrease, which lasted for more than 10 s, and hypopnoea was defined as at least 30% airflow decrease lasting for at least 10 s, which was related to a ≥3% oxygen desideration or an arousal. Total sleep time (TST), sleep period time (SPT), total sleep time spent with oxygen saturation below 90% (TST90%) and minimal oxygen saturation (minSatO_2_) were recorded, and apnoea–hypopnoea index (AHI), oxygen desaturation index (ODI) and arousal index (AI) were calculated. Obstructive sleep apnoea was defined as having an AHI ≥ 5/h.

### 2.3. SuPAR Measurements

Venous blood was taken into EDTA tubes. Within 30 minutes, blood samples were centrifuged at 4 °C for 10 min at 1500 rpm, and the plasma was stored at −80 °C until further analysis. Plasma suPAR levels were measured using a commercially available ELISA kit (ViroGates A/S, Birkerød, Denmark) as described previously [25]. The samples were measured in duplicates, and the mean concentration was used. The intra-assay coefficient of variation was 9 ± 11% with a lower limit of detection of 0.1 ng/mL. All suPAR concentrations were above the detection limit.

### 2.4. Statistical Analyses

Statistica 12 (StatSoft, Inc., Tulsa, OK, USA) was used for statistical analyses. The normality of the data was checked with the Kolmogorov–Smirnov test, which showed normal distribution for suPAR concentrations. Patient and control groups were compared with unpaired t-test, Mann–Whitney, Chi-square and Fisher tests. Plasma suPAR was related to clinical and demographic variables using linear and logistic regression and compared among different OSA severities with general mixed linear models. These analyses were repeated following adjustment for age, gender, body mass index (BMI), type of the sleep tests, anticoagulant and antithrombotic medications and GFR as well. To avoid the confounding effect of hypertension and diabetes, OSA and control groups were compared when subjects affected by these comorbidities were excluded. A *p* value <0.05 was considered significant. The suPAR results are presented as mean ± standard deviation with 95% confidence intervals.

The minimal sample size was estimated to detect differences in plasma suPAR levels between the OSA and control groups with an effect size of 0.80, power of 0.80 and alpha of 0.05 [26]. These numbers were based on a distribution of plasma suPAR values in control subjects [25]. Post-hoc sensitivity analyses ensured it was possible to detect correlations between suPAR and clinical variables within −0.23 and 0.23, minimal and maximal critical r values, statistical power of 0.80 and alpha of 0.05 [26].

The study was approved by the Semmelweis University Ethics Committee (TUKEB 30/2014 and 172/2018, approved on 26 October 2018) and was conducted according to the Declaration of Helsinki. Patients provided their written consent.

## 3. Results

### 3.1. Patient Characteristics

OSA was diagnosed in 53 cases (6 mild, 25 moderate and 22 severe; AHI 5–14.9/h, 15–29.9/h and ≥30/h, respectively). Patients with OSA had higher BMI, systolic (SBP) and diastolic blood pressure (DBP), AHI, ODI, SPT, TST, TST 90% and lower high density lipoprotein cholesterol (HDL-C) and MinSatO_2_ compared to controls (all *p* < 0.05, Table 1). In addition, patients with OSA tended to be older (*p* = 0.08) and sleepier (*p* = 0.05), and the prevalence of dyslipidaemia tended to be higher in OSA (*p* = 0.07, Table 1).

### 3.2. Circulating suPAR Results

There was no difference in plasma suPAR concentrations between the controls (2.088 ± 0.976/1.548–2.628/ ng/mL) and patients with OSA (2.198 ± 0.675/2.012–2.384/ ng/mL, unadjusted *p* = 0.62, *p* = 0.99 after adjustment, Figure 1). Similarly, there was no difference between mild (2.134 ± 0.799/1.295–2.974/ ng/mL), moderate (2.274 ± 0.597/2.028–2.521/ ng/mL) and severe patients (2.128 ± 0.744/1.798–2.458/ ng/mL, unadjusted *p* = 0.84, *p* = 0.78 after adjustment, Figure 2). In line with this, there was no relationship between plasma suPAR levels and AHI (*p* = 0.65), ODI (*p* = 0.58), TST90% (*p* = 0.35), minSatO_2_ (*p* = 0.16), AI (*p* = 0.38), TST (*p* = 0.60), SPT (*p* = 0.41) or ESS (*p* = 0.44). We noted two outliers in the control group. These subjects did not differ in their demographics or clinical characteristics from other controls. Excluding them from analyses resulted in tendency for higher suPAR levels in OSA (*p* = 0.050); however, after adjustment for covariates this difference was not significant (*p* = 0.259).

Plasma suPAR directly correlated with age when all subjects were analysed together (r = 0.33, *p* < 0.01), or when patients with OSA were investigated separately (r = 0.30, *p* = 0.02). However, when adjusting for covariates, these correlations were no longer significant (both *p*>0.05).

Plasma suPAR levels were higher in women when all subjects were analysed together (2.487 ± 0.683/2.241–2.733/ vs. 1.895 ± 0.692/2.221–2.713/ ng/mL, *p* < 0.01), in controls (2.539 ± 0.956/1.804–3.474/ vs. 1.411 ± 0.534/0.851–1.971/ ng/mL, *p* = 0.02) and in OSA (2.467 ± 0.568/2.221–2.713/ vs. 1.991 ± 0.686/1.735–2.247/ ng/mL, *p* < 0.01, Figure 3). These intergender differences remained significant even after adjustment for covariates. Due to the asymmetric gender distribution in the OSA and control groups, plasma suPAR levels were compared in control and OSA women and men separately. There was no difference in women (2.467 ± 0.568/2.221–2.713/ vs. 2.539 ± 0.956/1.804–3.474/ ng/mL, OSA vs. controls, *p* = 0.79). However, plasma suPAR tended to be higher in male patients with OSA (1.991 ± 0.686/1.735–2.247/ ng/mL, *n* = 30) compared to controls (1.411 ± 0.534/0.851–1.971/ ng/mL, *n* = 6, *p* = 0.059). Despite this potential signal, there was no relationship between AHI and suPAR levels in either men or women (both *p* > 0.05).

Patients with OSA who took anticoagulants had higher plasma suPAR levels (2.739 ± 0.547/2.164–3.313/ vs. 2.129 ± 0.663/1.934–2.323/ ng/mL, *p* = 0.03); however, this difference became insignificant after adjusting for covariates (*p*>0.05). None of the other correlations between plasma suPAR concentrations, demographics, clinical variables or comorbidities, such as hypertension, diabetes, dyslipidaemia or cardiovascular disease were significant in any of the studied groups (all *p*>0.05).

### 3.3. Plasma suPAR Results in Control and OSA Participants without Hypertension or Diabetes

There was no difference in plasma suPAR levels when controls without hypertension (1.903 ± 0.922/1.244–2.562/ ng/mL, *n* = 10) were compared to OSA patients without hypertension (2.110 ± 0.671/1.753–2.467/ ng/mL, *n* = 16, *p* = 0.51). Similarly, no difference was found between controls without diabetes (2.088 ± 0.976/1.548–2.628/ ng/mL, *n* = 15) and OSA patients without diabetes (2.203 ± 0.676/2.002-2.403/, *n* = 46, *p* = 0.61). In line with this, there was no difference when controls without hypertension and diabetes (1.903 ± 0.922/1.244–2.562/ ng/mL, *n* = 10) were compared to OSA patients without hypertension and diabetes (2.165 ± 0.656/1.802–2.528/ ng/mL, *n* = 15, *p* = 0.41).

## 4. Discussion

In the current study, we analysed plasma suPAR levels in OSA, but did not find any difference compared to controls, nor did suPAR concentrations correlate with disease severity. This implies that suPAR may not play a significant role in the pathophysiology of OSA; however, due to the small number of controls and the significant gender effect on suPAR levels, our results must be interpreted carefully.

Obstructive sleep apnoea is associated with heightened systemic inflammation, theoretically contributing to higher uPAR expression [11,13,14,15]. However, the cleavage of uPAR may be slower in OSA due to decreased levels of uPA [8] and plasmin [5,6,23]. The expression of uPA is induced by female sexual hormones [20] and the proto-oncogenic survivin [27], which presented decreased expressions in OSA [3,28]. Plasmin is formed by plasminogen, and this reaction is blocked by plasminogen activator-inhibitors (PAI), shown to be upregulated in OSA [5,6,23]. In addition, OSA is associated with decreased levels of transforming growth factor-β [8], an inducer of uPAR transcription [29,30]. These studies suggest that although uPAR expression may be upregulated by systemic inflammation in OSA [11,13,14,15]; this is counterbalanced by the reduced cleavage.

It has been shown that plasma suPAR levels are higher in women [20] and related to BMI and waist circumference only in females [20]. In addition, plasma suPAR levels were prognostic for cardiovascular events only in women [19] and more strongly related to coronary artery calcification in women than in men [31]. Our results are in line with the previous findings [20], namely that suPAR was higher in women in both OSA and controls. A potential reason for the gender differences is that uPA is released upon stimulation by progesterone and oestradiol [32] resulting in higher uPAR cleavage. Female sexual hormones are protective in OSA [33], contributing to male predominance in sleep apnoea [33,34]. To exclude this effect, analyses were performed after adjustment for gender and suPAR was compared between OSA and controls in women and men separately. Although there was a tendency for higher suPAR levels in men in OSA, there was no relationship with OSA severity in males. Of note, the number of men in the control group was small, and these analyses were underpowered. Nevertheless, this difference could be a potential signal which should be investigated in further studies. We believe that our current results would provide basis for further study design. In line with the previous findings [35], plasma suPAR levels were directly related to age; however, this correlation disappeared after adjustment for covariates. Although higher suPAR levels were associated with obesity [21], this has not been confirmed by the current study.

Our study has limitations. First, the sample size, especially in the control group, was low. This could have potentially led to type II error, especially due to significant difference in age, gender and comorbidity distribution. To avoid this, our analyses were adjusted on potential confounders. The plasma suPAR levels were not different between patients with OSA and controls either in unadjusted or adjusted comparisons. Still, our results should be interpreted carefully, especially considering the exclusion of the two outliers which resulted in differences between the two groups. The sample size calculations were based on our previous study [25], showing higher suPAR levels in COPD. Although the number of participants may seem low, it may not be the likely reason for the lack of differences between OSA and controls considering the wide overlap of suPAR values between the two groups and the lack of significant relationship between markers of OSA severity and suPAR levels. In line with this, a second limitation is the unbalanced proportion of comorbidities in the OSA and control groups.

Elevated suPAR levels are associated with cardiovascular disease and diabetes [36]. OSA represents a risk for cardio metabolic disease [22], which was reflected in the asymmetric proportion of comorbidities in the OSA and control groups. However, we did not find any relationship between plasma suPAR concentrations and comorbidities. To further evaluate this, we performed additional analyses in participants without hypertension or diabetes. We did not find any difference in plasma suPAR values between controls and patients with OSA in non-hypertensive or nondiabetic volunteers. Of note, the study has not been powered to address this question. We believe our results could provide a basis to design further studies involving groups balanced on the profile of comorbidities. The third limitation is that although patients represented a large range of OSA severity, in average, they were minimally symptomatic. It has recently been reported that patients with OSA and excessive daytime sleepiness have a higher risk for cardiovascular disease [37]. Inclusion of more symptomatic patients in studies investigating systemic inflammation is therefore warranted. The fourth limitation is the significant gender-effect which has been discussed above. The strengths of the study include the application of objective sleep tests, detailed characterisation of the studied population and robust methodology for plasma suPAR measurement.

Only one study has examined suPAR in possible OSA. Von Kanel et al. divided 329 South African teachers based on their response to the Berlin questionnaire and/or neck circumference into a high-risk and low-risk OSA group. Most notably, no objective sleep study has been performed. Although the levels of fibrinogen and PAI-1 were elevated together with slower clot lysis time, there was only a tendency for higher suPAR levels in the high-risk group [23]. The Berlin questionnaire is a moderately sensitive, but not specific screening tool for OSA [38]; therefore, these results must be interpreted carefully. Nevertheless, the previous [23] and the current findings indicate that hyper-coagulation in OSA is driven by high fibrin formation, reduced plasminogen activation by increased PAI-1 and lower uPA without a significant difference in the uPAR signalling.

## 5. Conclusions

In conclusion, we did not find altered plasma suPAR levels in patients with OSA vs. controls. Our results suggest that this molecule does not play a significant role in hyper coagulation and accelerated systemic inflammation in OSA and cannot be applied as a readout signal for these pathophysiological processes. However, the significant gender differences are noteworthy and must be considered when designing future studies with suPAR.

## Figures and Tables

**Figure 1 medicina-56-00077-f001:**
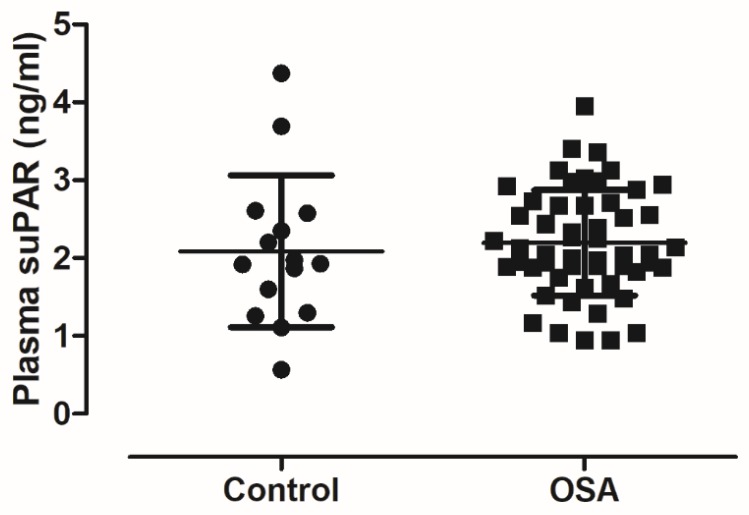
Comparison of plasma soluble urokinase-type plasminogen activator receptor (suPAR) levels between patients with OSA and controls. There was no difference between the two groups in plasma suPAR levels (*p* = 0.62). Mean ± standard deviation is presented.

**Figure 2 medicina-56-00077-f002:**
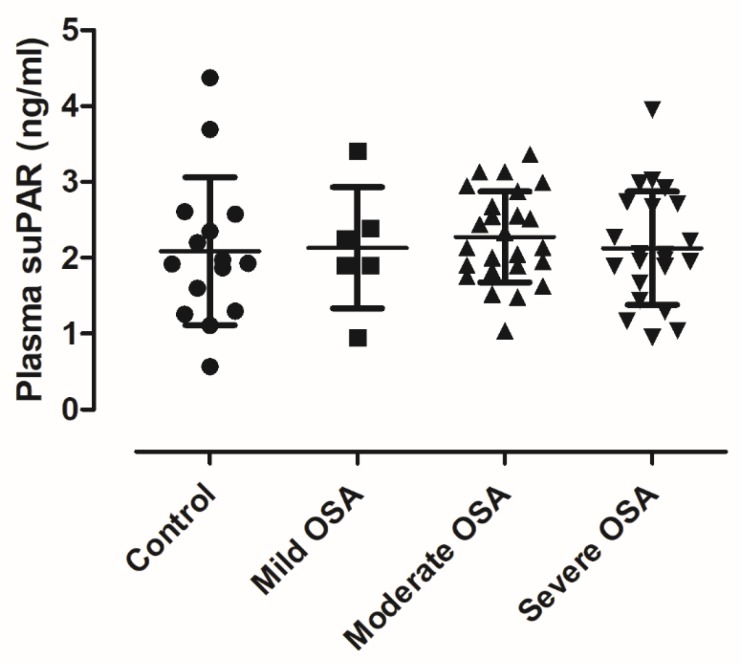
Comparison of plasma suPAR levels among different disease severities. There was no difference among the groups in plasma suPAR levels (*p* = 0.87). Mean ± standard deviation is presented.

**Figure 3 medicina-56-00077-f003:**
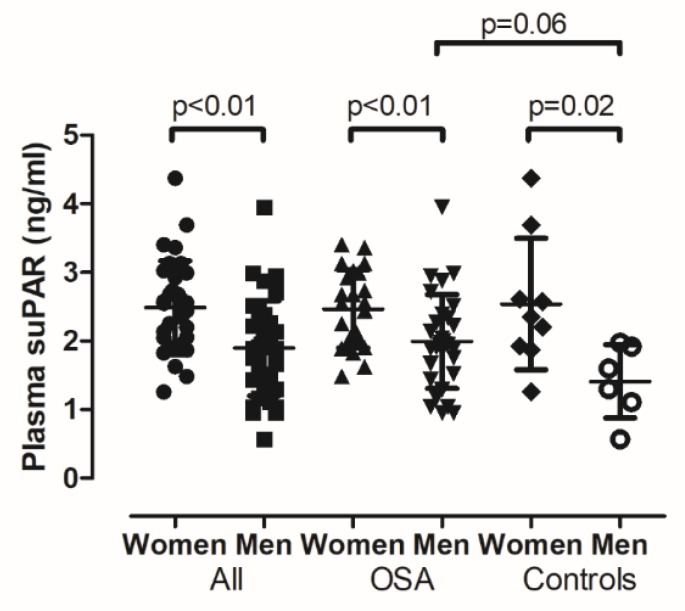
Comparison of plasma suPAR levels between women and men. Plasma suPAR levels were higher in women in patients with OSA, in controls and when the subjects were analysed together. Mean ± standard deviation is presented.

**Table 1 medicina-56-00077-t001:** Subjects’ characteristics *.

	Control (*n* = 15)	OSA (*n* = 53)	Total (*n* = 68)	*p*
Age (years)	48/31–62/	59/49–64/	54.6/44–64/	0.08
Male (*n*, %)	6, 40%	30, 57%	36, 53%	0.26
BMI (kg/m^2^)	24.6±4.58	32.37±5.66	30.66±6.31	**<0.01**
Hypertension (%)	33	70	62	**0.01**
Diabetes (%)	0	13	10	0.14
Dyslipidaemia (%)	27	53	47	0.07
Cardiovascular disease (%)	13	11	12	0.83
Cardiac arrhythmia (%)	13	26	24	0.29
Smokers (%)	20	23	22	0.83
Subjects taking anticoagulants (%)	0	11	10	0.32
Subjects taking antithrombotic drugs (%)	13	11	12	1.00
SBP (mmHg)	120/118–131/	138/130–150/	132/120–140/	**<0.01**
DBP (mmHg)	70/70–75/	85/76–90/	80/70–90/	**<0.01**
Creatinine (mmol/L)	77.5/66.3–83.5/	66/61–79/	70.0/62.0–82.5/	0.17
GFR (mL/min/1.73m^2^)	82.26±15.17	88.37±17.91	86.75±17.3	0.26
CRP (mg/L)	2.34/0.92–3.95/	3.0/1.36–4.65/	3.0/1.33–4.48/	0.23
Glucose (mmol/L)	4.65/4.1–5.3/	5.2/4.8–6.1/	5.1/4.7–5.7/	0.05
Cholesterol (mmol/L)	5.91±1.04	5.33±1.13	5.46±1.13	0.08
HDL-C (mmol/L)	1.84/1.58–2.34/	1.28/1.06–1.62/	1.36/1.17–1.82/	**<0.01**
LDL-C (mmol/L)	3.41±0.89	3.16±1.02	3.22±0.99	0.39
Triglyceride (mmol/L)	1.23/0.94–1.32/	1.33/1.07–1.97/	1.28/1.04–1.92/	0.10
Lipoprotein (a) (mmol/L)	0.2/0.0–0.4/	0.17/0.10–0.49/	0.18/0.07–0.44/	0.56
AHI (1/h)	1.6/1.0–2.4/	27.5/18.55–42.75/	21.89/8.9–40.63/	**<0.01**
ODI (1/h)	0.9/0.4–1.7/	23.2/14.1–38.95/	19.65/6.2–32/	**<0.01**
SPT (min)	395.19±36.1	439.93±72.82	427.81±67.62	**0.04**
TST (min)	398/319–409.5/	428/384.5–452/	410.5/363.25–440.75/	**<0.01**
TST90% (%)	0.0/0.0–0.1/	6/1.7–12.5/	3.85/0.4–11.6/	**<0.01**
Minimal oxygen saturation (%)	91/88.25–92.75/	82.5/76.8–85/	83/77.8–88/	**<0.01**
ESS	5.0/2.75–7.0/	7/5–10/	6/4–8/ 6.6±3.68	0.05

* Data are presented as mean ± standard deviation or median/25%–75% percentile/. Significant differences are highlighted in bold. AHI—apnoea–hypopnoea index, BMI—body mass index, DBP—diastolic blood pressure, CRP—C-reactive protein, ESS—Epworth Sleepiness Scale, GFR—glomerular filtration rate, HDL-C—high density lipoprotein cholesterol, LDL-C—low density lipoprotein cholesterol, ODI—oxygen desaturation index, SBP—systolic blood pressure, SPT—sleep period time, TST—total sleep time, TST90%—total sleep time spent with oxygen saturation below 90%.

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
