# Peer review of "Circulating Soluble Urokinase-Type Plasminogen Activator Receptor in Obstructive Sleep Apnoea"

_medicina, 2020, doi:10.3390/medicina56020077_

Round 1

Reviewer 1 Report

This is nicely done and well-written. I congratulate the authors on an excellent project.

However, the limitations of this study should be fully declared. For instance, one drawback of this study is the small sample size. OSA is a male-predominant disorder, and the male-to-female sex ratio has been reported as 1.67 to 3.25:1 (Sleep apnea and risk of vertigo: A nationwide population-based cohort study, Tsai MS et al, 2018 Laryngoscope). There are only six male controls in this study. The above limitations should be mentioned in the discussion.

Author Response

Comment: However, the limitations of this study should be fully declared. For instance, one drawback of this study is the small sample size. OSA is a male-predominant disorder, and the male-to-female sex ratio has been reported as 1.67 to 3.25:1 (Sleep apnea and risk of vertigo: A nationwide population-based cohort study, Tsai MS et al, 2018 Laryngoscope). There are only six male controls in this study. The above limitations should be mentioned in the discussion.

Response: Thank you for your comments. We agree that the biggest drawback is the small sample size and the unbalanced distribution of genders in the studied population. Thank you for the suggestion, we added this reference to the manuscript. We expanded the limitations paragraph in the Discussion and toned down the conclusions.

Reviewer 2 Report

Bocskei et al have performed a study to analyze the relationship between uPAR  (urokinase-type plasminogen activator receptor) concentration and obstructive sleep apnea using a case-control study. They concluded that there is no relationship between uPAR and OSA diagnosis or severity, so this molecule does not play a significant role in the pathophysiology of OSA. However they found higher levels of uPAR in women compared with men.

I have some comments.

Please, the objective should be clearly stated. The number of patients and (especially) controls included are small. I suggest the authors tone down the conclusions. Perhaps could be better to say that the results SUGGEST that plasma suPAR does not play…. Were there differences depending on the sleep diagnostic test used? Statistical analysis. I see that the authors have calculated a sample size, however I think that the number of controls is very limited (only 15 patients). Plasma suPAR was expressed as mean (SD). Did it follow a normal distribution? With such as small number of patients the inclusion of confounders is not easy under a statistical point of view. I think that the better statistical test to analyze this study is a propensity score matching test using as a confounders at least BMI, gender, age and those drugs associated with changes in the coagulability state. It is curious that the OSA patients recruited from a sleep lab have this low Epworth scale 7 (5-10) My main concern is the limited number of controls. Seeing the figure 1 I see that there are even some outliers…. The authors should clearly state in the discussion section a paragraph with the strengths and the limitations of the study. Some important references should be added, for example a review recently published in the European Respiratory Journal by Garcia-Ortega et al (Eur Respir J 2019; 53: 1800893) The analysis of the difference between men and women is clearly underpowered. For example there are only 6 men in the control group (see figure 3)

Author Response

Comment: Please, the objective should be clearly stated.

Response: The study objective was to compare plasma suPAR levels in OSA and controls and to correlate them to indices of OSA severity. We clarified this in the last paragraph of Introduction in the revised version.

Comment: The number of patients and (especially) controls included are small. I suggest the authors tone down the conclusions. Perhaps could be better to say that the results SUGGEST that plasma suPAR does not play….

Response: Thank you for your comment. We expanded the limitations section of the Discussion and toned down the conclusions in the abstract and main text.

Comment: Were there differences depending on the sleep diagnostic test used?

Response: There was no difference in plasma suPAR levels depending on the diagnostic test used. Furthermore, we adjusted our calculations for this potential confounder. Please see methods and results.

Comment: Statistical analysis. I see that the authors have calculated a sample size, however I think that the number of controls is very limited (only 15 patients). Plasma suPAR was expressed as mean (SD). Did it follow a normal distribution? With such as small number of patients the inclusion of confounders is not easy under a statistical point of view. I think that the better statistical test to analyze this study is a propensity score matching test using as a confounders at least BMI, gender, age and those drugs associated with changes in the coagulability state.

Response: We agree with the reviewer regarding the low number of subjects and hence we toned down the conclusions. The subject number was based on our previous publication on suPAR in COPD (Bocskei et al. Lung, 2019). With this sample size we could also demonstrate (the lack of) correlations between suPAR and indices of OSA severity with correlation coefficients greater than 0.23. The data showed normal distribution. We appreciate the reviewer comment to use propensity score matching test. This is a test developed for interventional trials but its application in observational settings is debated (https://gking.harvard.edu/files/gking/files/psnot.pdf). To adjust for this covariates we applied general mixed linear model and used age, gender, BMI, medications, sleep tests and GFR as covariates. There was no difference after adjustment between the OSA and control groups (p=0.99) or along different severity groups (p=0.78).

Comment: It is curious that the OSA patients recruited from a sleep lab have this low Epworth scale 7 (5-10)

Response: The volunteers were selected from subjects referred to the Sleep Lab due to suspected OSA (i.e. snoring, daytime tiredness, obesity and comorbidities). The subjects did not necessarily complain about sleepiness. Of note, sleepiness was measured with a subjective questionnaire and the correlation between OSA severity and sleepiness would be more evident using PVT, MSLT or MWT. However, patients with excessive sleepiness have higher risk for cardiovascular disease (Mazzotti DR et al. Am J Respir Crit Care Med. 2019). Minimally symptomatic patients with OSA could be the potential reason for the lack of differences in suPAR in the current study.  

Comment: My main concern is the limited number of controls. Seeing the figure 1 I see that there are even some outliers…. The authors should clearly state in the discussion section a paragraph with the strengths and the limitations of the study.

Response: The two outliers in the control group were female, overweight (both BMI were 27) non-smokers and one of them has been diagnosed with hypertension. None of them took anticoagulants or anti-thrombotic drugs. When excluding these two subjects, there was a tendency for higher suPAR values in OSA (p=0.050), however after adjustment on covariates this difference was not significant. We expanded the 4th paragraph of the Discussion with strengths and limitations of the study.

Comment: Some important references should be added, for example a review recently published in the European Respiratory Journal by Garcia-Ortega et al (Eur Respir J 2019; 53: 1800893).

Response: Thank you for your comment. Three references, including the suggested one have been added to the manuscript.

Comment: The analysis of the difference between men and women is clearly underpowered. For example there are only 6 men in the control group (see figure 3)

Response: We agree with the reviewer. We expanded the 3rd paragraph of Discussion and acknowledged this problem.

Round 2

Reviewer 2 Report

The authors have properly addressed almost all my concerns